# Structural and Biochemical Characterization of a Dye-Decolorizing Peroxidase from *Dictyostelium discoideum*

**DOI:** 10.3390/ijms22126265

**Published:** 2021-06-10

**Authors:** Amrita Rai, Johann P. Klare, Patrick Y. A. Reinke, Felix Englmaier, Jörg Fohrer, Roman Fedorov, Manuel H. Taft, Igor Chizhov, Ute Curth, Oliver Plettenburg, Dietmar J. Manstein

**Affiliations:** 1Institute for Biophysical Chemistry, Hannover Medical School, Fritz Hartmann Centre for Medical Research Carl Neuberg Str. 1, D-30625 Hannover, Germany; Amrita.Rai@mpi-dortmund.mpg.de (A.R.); patrick.reinke@desy.de (P.Y.A.R.); Taft.Manuel@mh-hannover.de (M.H.T.); chizhov.igor@mh-hannover.de (I.C.); curth.ute@mh-hannover.de (U.C.); 2Department of Structural Biochemistry, Max Planck Institute of Molecular Physiology, D-44227 Dortmund, Germany; 3Department of Physics, University of Osnabrueck, Barbarastrasse 7, D-49076 Osnabrück, Germany; jklare@uni-osnabrueck.de; 4Division for Structural Biochemistry, Hannover Medical School, Carl Neuberg Str. 1, D-30625 Hannover, Germany; Fedorov.Roman@mh-hannover.de; 5Center for Free-Electron Laser Science, German Electron Synchrotron (DESY), Notkestr. 85, D-22607 Hamburg, Germany; 6Institute of Medicinal Chemistry, Helmholtz Zentrum München (GmbH), German Research Center for Environmental Health, Ingolstädter Landstraße 1, D-85764 Neuherberg, Germany; felix.englmaier@helmholtz-muenchen.de (F.E.); oliver.plettenburg@oci.uni-hannover.de (O.P.); 7Center of Biomolecular Drug Research (BMWZ), Institute of Organic Chemistry, Leibniz University Hannover, Schneiderberg 1b, D-30167 Hannover, Germany; joerg.fohrer@tu-darmstadt.de; 8NMR Department of the Department of Chemistry, Technical University Darmstadt, Clemens Schöpf Institute for Organic Chemistry and Biochemistry, Alarich-Weiss-Strasse 4, D-64287 Darmstadt, Germany; 9RESiST, Cluster of Excellence 2155, Medizinische Hochschule Hannover, D-30625 Hannover, Germany

**Keywords:** dye-decolorizing-type peroxidase, heme peroxidases, lignin degradation, *Dictyostelium discoideum*, B-type DyP, electron paramagnetic resonance (EPR) spectroscopy, compound I, enzyme kinetics, crystal structure, long-range electron transfer

## Abstract

A novel cytoplasmic dye-decolorizing peroxidase from *Dictyostelium discoideum* was investigated that oxidizes anthraquinone dyes, lignin model compounds, and general peroxidase substrates such as ABTS efficiently. Unlike related enzymes, an aspartate residue replaces the first glycine of the conserved GXXDG motif in *Dictyostelium* DyPA. In solution, *Dictyostelium* DyPA exists as a stable dimer with the side chain of Asp146 contributing to the stabilization of the dimer interface by extending the hydrogen bond network connecting two monomers. To gain mechanistic insights, we solved the *Dictyostelium* DyPA structures in the absence of substrate as well as in the presence of potassium cyanide and veratryl alcohol to 1.7, 1.85, and 1.6 Å resolution, respectively. The active site of *Dictyostelium* DyPA has a hexa-coordinated heme iron with a histidine residue at the proximal axial position and either an activated oxygen or CN^−^ molecule at the distal axial position. Asp149 is in an optimal conformation to accept a proton from H_2_O_2_ during the formation of compound I. Two potential distal solvent channels and a conserved shallow pocket leading to the heme molecule were found in *Dictyostelium* DyPA. Further, we identified two substrate-binding pockets per monomer in *Dictyostelium* DyPA at the dimer interface. Long-range electron transfer pathways associated with a hydrogen-bonding network that connects the substrate-binding sites with the heme moiety are described.

## 1. Introduction

The social amoeba *Dictyostelium discoideum* is unusual among eukaryotes in having both unicellular and multicellular stages [1]. *Dictyostelium discoideum* cells are frequently found as an abundant component of the microflora in the upper layer of soil and on decaying organic material [2]. Here, they play an important role as phagotrophic bacterivores in the maintenance of balanced bacterial populations [3]. The peroxidase database RedoxiBase [4] suggests that *Dictyostelium discoideum* produces a single dye-decolorizing peroxidase (DyP). DyPs have distinctive catalytic properties, among them a uniquely broad substrate acceptance profile that includes diverse organic dyes [5]. In addition to anthraquinone-based dyes and lignin model compounds, they have been shown to degrade 2, 6-dimethoxyphenol, guiaiacol, pyrogallol, azo dyes, ascorbic acid, ß-carotene, and phenolic compounds [6,7,8,9]. Furthermore, DypB from *Rhodococcus jostii* RHA1, *Amycolatopsis* sp. 75iv2 DyP2, *Pseudomonas fluorescens* DyP1B, and DyPs from *Pleurotus ostreatus* have been shown to oxidize Mn^2+^ [10,11,12,13]. Phylogenetically, the DyP superfamily can be subdivided into five different classes [14]. Class A consists of TAT-dependent secreted enzymes, while class B and C include cytoplasmic enzymes that are produced in bacteria and lower eukaryotes. Class B and C proteins are produced without N-terminal extension. Class D enzymes are primarily fungal DyPs that typically have an N-terminal presequence that is processed during maturation. Class E enzymes are involved in stress-response pathways, and the expression of *dyp* genes was shown to be upregulated in archaea and several pathogenic bacteria under oxidative stress conditions. Class E enzymes are the least characterized DyPs [14]. Crystal structures of DyP family members reveal two domains, each one adopting an α + β ferredoxin-like fold, which makes them structurally distinct from other peroxidase superfamily members [15]. An alternative, structure-based classification system subdivides DyPs only into three classes. Here, class I (Intermediate) corresponds to former class A, class P (Primitive) to class B, and class V (Advanced) to former classes C and D [16].

The catalytic mechanism of DyPs resembles that of plant-type peroxidases [15]. The resting ferric enzyme reacts with H_2_O_2_ to yield compound I, a high-valent intermediate [Fe^IV^=O Por^•^]^+^. Loss of one electron from compound I in the presence of reducing substrate leads to the formation of compound II [Fe^IV^=O] which in turn decays into the resting state Fe^III^ peroxidase after reacting with a second equivalent of the reducing substrate [15]. Although residues on the distal face of the heme are different in plant-type peroxidases and DyPs, the heme is similarly ligated by a proximal histidine. DyPs have a conserved aspartate and arginine on the distal face, while a catalytic histidine is present in the plant-type peroxidases [17].

Here, we describe the biochemical and structural properties of *Dictyostelium* DyPA, the first DyP for an organism from the order Dictyosteliales. We tested the catalytic activity of *Dictyostelium* DyPA with a range of different organic substrates and characterized its ferric-heme microenvironment and the formation of catalytic intermediates by UV-Vis, EPR, and time-resolved stopped-flow spectroscopy. X-ray structures of *Dictyostelium* DyPA in complex with activated oxygen alone and together with veratryl alcohol, as well as the structure of the complex with the competitive inhibitor cyanide, provide detailed insight into the substrate access channel, active site residues, and movement of the DXXDG motif during the formation of compound I, with Asp149 functioning as an acid-base catalyst at low pH [7,17]. Asp146, which in *Dictyostelium* DyPA replaces the highly conserved glycine residue that is present in the first position of the motif in other Dyp family members (Figure 1A), contributes to the hydrogen-bond network between the DyPA monomers. Moreover, we describe long-range electron transfer pathways that appear to connect the ferric-heme center of *Dictyostelium* DyPA with surface-bound substrates.

## 2. Results and Discussion

### 2.1. Localization of Dictyostelium DyPA

The coding sequence of the *Dictyostelium dypA* gene (GenBank: EAL70759.1) consists of 921 base pairs and contains no introns. According to DictyBase, two copies of the *dypA* gene (DDB_G0273083 and DDB_G0273789) are present on chromosome 2 of *Dictyostelium discoideum* strains AX3 and AX4 [18]. The identical *dypA* gene copies encode a 306 amino acid (Mr = 34,965.6 Da) peroxidase, which like other DyPs has a catalytic aspartic acid and arginine over the heme plane (distal). The maximum-likelihood tree constructed by Ahmad and co-workers, which is derived from structure-based sequence alignments of DyP superfamily proteins, indicates that *Dictyostelium* DyPA belongs to the dye-decolorizing subfamily B (or class P) [16,19]. *Dictyostelium* DyPA is 46.0% identical in amino acid sequence to YfeX from *Escherichia coli* O157, 39% to *Vc*DyP from *Vibrio chlorae*, 37.7% to TyrA from *Shewanella oneidensis*, 33.6% to DypB from *Rhodococcus jostii* RHA1, and 31.1% to *Bt*Dyp from *Bacteroides thetaitaomicron* VPI-5482 (Appendix A). DyP-type peroxidases contain a conserved GXXDG motif in their primary sequence, which forms part of the heme-binding region. This motif is conserved in all reported DyPs, but in the case of *Dictyostelium* DyPA, the first glycine residue of this motif is replaced by an aspartate residue (DFIDG) (Figure 1A). No signal peptide or transmembrane regions are present in *Dictyostelium* DyPA, suggesting that the protein is neither secreted nor a membrane-bound protein. To check the cellular localization of the protein, we generated and overproduced N-terminal and C-terminal EYFP-fused *Dictyostelium* DyPA constructs in *Dictyostelium discoideum* cells. Confocal images of overproducing cells show the cytoplasmic localization of both N- and C-terminal tagged *Dictyostelium* DyPA (Figure 1B).

For biochemical characterization, recombinant *Dictyostelium* DyPA was overproduced and purified from *Escherichia coli* cells, as described previously [20]. Purified *Dictyostelium* DyPA protein (Apo-form) was faintly yellow with a very small Soret peak at 410 nm and a Reinheitszahl (R_z_ value Abs_Soret_/Abs_280_) of 0.13, indicating the presence of a small, substoichiometric amount of heme. Heme reconstitution was performed by adding hemin chloride in a 2:1 molar excess to the apo-protein, followed by size exclusion chromatography to remove any unbound heme. Heme reconstituted *Dictyostelium* DyPA was used throughout the study unless otherwise stated. The heme reconstituted protein displays an R_z_ of 2.0 and has a Soret band at 400 nm as well as charge transfer (CT) and Q bands at 638 and 506 nm, respectively, indicating a typical high-spin ferric-heme absorption spectrum. The heme content determined by the hemochromogen method corresponds to 0.91 mole heme per mole of reconstituted *Dictyostelium* DyPA. UV-visible absorption spectroscopy indicates that the heme microenvironment is sensitive to changes in pH. The peak value of the Soret band corresponds to 402 nm in the pH range 4–5, whereas at a higher pH (6–9), it is shifted to 400 nm. The R_z_ value did not change over the pH range of 6.0–9.0 and was 1.85 and 1.91 at pH 4.0 and 5.0, respectively. Soret band broadening was observed at pH 3.0 (Figure 1C).

### 2.2. Analysis of the Oligomerization State of Dictyostelium DyPA by Analytical Ultracentrifugation

Several oligomeric states of DyPs have been reported so far, ranging from monomers to hexamers [15]. To examine the exact oligomeric nature of *Dictyostelium* DyPA, we performed sedimentation velocity experiments in the analytical ultracentrifuge (SV-AUC). Protein concentrations from 2.1 to 33.6 µM were used. Sedimentation coefficient distributions calculated with the program SEDFIT [21] showed that, independent of the protein concentration used, about 90% of *Dictyostelium* DyPA sediments with an s_20,w_ of 4.8 S (see Figure 1D). From the sedimentation coefficient and the diffusion broadening of the sedimenting boundary, a molar mass of 67 kg/mol was obtained by the continuous c(s) distribution model in SEDFIT. Since the molar mass of the *Dictyostelium* DyPA monomer, as calculated from amino acid composition, is 35 kg/mol, *Dictyostelium* DyPA exists predominately as a dimer in solution. Compared to an unhydrated spherical dimer, a frictional ratio of 1.25 can be calculated from the sedimentation coefficient. Frictional ratios of spherical hydrated proteins are typically in the range of 1.1–1.2 [22]. Therefore, the shape of the *Dictyostelium* DyPA dimer appears to deviate only slightly from that of a perfect sphere.

Independent of *Dictyostelium* DyPA concentration, approximately 10% of the protein sedimented with an s_20,w_ of 3.0 S. Such an s-value would be expected for monomeric *Dictyostelium* DyPA, with a similar frictional ratio as observed for the dimer. Since the fraction of this species did not change when the protein concentration was varied by a factor of 16, no *Dictyostelium* DyPA monomer-dimer equilibrium appears to exist in the concentration range examined. In our *Dictyostelium* DyPA preparation, the heme saturation was greater than 90%. Therefore, we wanted to investigate the effect of heme on *Dictyostelium* DyPA dimerization. SV-AUC analysis of apo-*Dictyostelium* DyPA containing a heme saturation of only 7%, showed a slight decrease in the s_20,w_ of the main species to 4.3 S (Figure 1D) and a molar mass of 63 kg/mol. Thus, even in the absence of heme, *Dictyostelium* DyPA forms dimers. However, the increase in the frictional ratio to 1.37 indicates that *Dictyostelium* DyPA dimers are less compact in the absence of bound heme. Interestingly, apo- *Dictyostelium* DyPA also contains approximately 10% of a slower sedimenting species (s_20,w_ = 2.7 S). Therefore, both at high and low heme saturation, there exists a slower sedimenting species, which lacks the competence for dimer formation.

### 2.3. Absorption Spectra of Dictyostelium DyPA in the Presence of Peroxide or Cyanide and the Formation of Compound I

To investigate the formation of reaction intermediates, *Dictyostelium* DyPA was mixed with H_2_O_2_, and absorption spectra were recorded. The addition of 1 equivalent of H_2_O_2_ to *Dictyostelium* DyPA at pH 8.0 resulted in broadening and a slight blue-shift of the Soret peak to 396 nm, a prominent shoulder at 340 nm, and a CT band shifted from 635 to 648 nm. A broad hyperchromatic region was observed between 500 and 636 nm (Figure 1E). The *Dictyostelium* DyPA spectrum is similar to reported plant peroxidases or other DyPs compound I [Fe^IV^=O Por^•^]^+^ reaction intermediates [10,15,23,24]. Since earlier studies have shown that peroxidase–CN^−^ complexes are a good and stable mimic of the peroxidase-H_2_O_2_ bound state [25], we recorded the UV-Vis spectra of *Dictyostelium* DyPA at pH 8.0 in the presence of KCN. The addition of KCN to *Dictyostelium* DyPA shifts the Soret band from 400 to 419 nm. The CT band at 635 nm disappeared, and the Q band shifted from 506 nm to a broader band at 534 nm with a shoulder at 564 nm. Thus, the spectral changes observed for *Dictyostelium* DyPA-CN^−^ are similar to those reported for other peroxidases and in particular for the *Arthromyces ramosus* and *Geotrichum candidum Dy*P-CN^−^ complexes [26,27]. The absorption spectrum suggests that the binding of CN^−^ leads to a change in the electronic state of iron from high spin to low spin (Figure 1F).

As the speed of the reaction between DyPs and H_2_O_2_ is very fast, the rate of the formation of compound I as an intermediate was investigated using stopped-flow measurements. Rapid mixing of *Dictyostelium* DyPA with H_2_O_2_ led to the decay of the Soret peak and the appearance of bands characteristic for compound I. The decay of the Soret peak can be described by a single exponential equation. The second-order rate constant for the formation of compound I (2.08 ± 0.16 × 10^6^ M^−1^s^−1^) was obtained from the slope of a plot of the observed rate constants against the H_2_O_2_ concentration (Figure 1G). The rate constant is approximately 10-fold higher than for *Rj*DypB [10] and 10-fold lower than for plant peroxidases such as horseradish peroxidase (1.7 ± 0.1 × 10^7^ M^−1^s^−1^) [28]. *Dictyostelium* DyPA compound I is relatively stable (~10 min) and does not decay into compound II but rather returns to the resting ferric state (Appendix A). Similar observations were made with *Rj*DypB and *Bad*DyP (class B/D) [10,17]. Class A type DyPs such as *Rj*DypA favor the formation of compound II in the presence of H_2_O_2_, without detectable accumulation of compound I [10].

### 2.4. Electron Paramagnetic Resonance Spectroscopy

Figure 2A (top) shows the low-temperature (6 K) 9.4 GHz EPR spectrum of *Dictyostelium* DyPA. It consists primarily of high-spin ferric species (*S* = 5/2) characterized by two resonances at *g*^eff^_⊥_ ≈ 6 and *g*^eff^_||_ ≈ 2, and a minor contribution of a low-spin ferric form responsible for the weak signals observed at *g* = 2.81 and *g* = 2.28. As expected, the spectrum is similar to previously reported EPR spectra of other peroxidases such as KatG [29,30] and especially those recently reported for DypA and DypB from *Rhodococcus jostii* RHA1 [10]. Simulation of the EPR spectrum (Figure 2A) revealed the presence of (i) two rhombically distorted (*g_x_* ≠ *g_y_*) but near-axial high-spin species, HS 1 and HS 2, (ii) an axial high spin species, HS 3, and (iii) a small amount (4%) of low-spin heme, LS. The simulation parameters are given in Table 1. Further analysis of the rhombic high spin components in terms of the zero-field splitting parameters *E/D* using the absolute difference in *g*_⊥_ values (*g_x_*–*g_y_*) [31] revealed very similar values of *E/D* ≈ 0.021 corresponding to ~6.3% rhombicity for both species. This value is similar to those found for the two *Rhodococcus. jostii* RHA1 enzymes (*Rj*DypA: 5.44%, *Rj*DypB: 4.06%) [10], indicating similar coordination microenvironments for the heme iron in the paralogs. An axial (g_x_ = *g_y_* = *g*_⊥_) species has not been found for the *Rhodococcus jostii* RHA1 enzymes but was found for the *Synechocystis* KatG with very similar *g*-values (*g*_⊥_ = 5.93, *g*_||_ = 1.99) [30]. Multiple heme conformations such as those observed here were previously reported for the majority of peroxidases investigated so far [10,29,30,32,33].

Upon reaction of *Dictyostelium* DyPA (250 µM) with excess H_2_O_2_ (3 mM) for ~5 s on ice, leading to a color change of the enzyme from brown to green, an asymmetric spectral feature centered at *g* ≈ 2.00 indicative of an isolated organic (protein-based) radical is observed (Figure 2B). Interestingly, the hyperfine structure of the *g* ≈ 2.00 spectral feature resembles more what was observed for catalase-peroxidase from *Synechocystis* PCC6803, which has been accounted for by the formation of tyrosine (Tyr^•^) and tryptophan radicals (Trp^•^) but is clearly distinct from the spectral features observed with *Rj*DypB [10]. Temperature-dependency studies in the range from 6–40 K (Figure 3A) revealed the presence of at least two species. One set of resonances appeared to be temperature independent, whereas other resonances disappeared with increasing temperature. As for *Rj*DypB, the same sample recorded under nonsaturating conditions at 40 K (inset in Figure 3A) showed minor contributions from noncoupled protein-based radical(s). The resonances at g ≈ 6 appeared to be largely altered toward at least two axial species with reduced intensity (Figure 2B). Although the persistence of these ferric components can be explained in part by nonreactive enzyme, the absence of the rhombically distorted species identified for the unreacted enzyme and the appearance of at least one additional largely axial form suggest that (i) the native enzyme does not make a significant contribution to the EPR spectrum; (ii) the newly formed axial component(s) either characterize an additional intermediate in *Dictyostelium* DyPA’s enzymatic cycle or indicate that (an) organic radical(s) is (are) accompanied by a ferric rather than a Fe^4+^ oxoferryl center in a fraction of the enzyme. The low-spin species LS, characterized by effective *g*-values of 2.81, 2.28, and 1.99 disappeared almost entirely (spurious amounts of LS are deducible from the inset in Figure 2B, marked by stars) upon reaction with H_2_O_2_.

To gain further insights into the nature and temperature dependency of the organic radicals, we calculated difference spectra from the temperature dependency data (Figure 3B). All components visible only at T < 20 K are reflected in the (6–20 K) difference spectrum shown at the top. The (6–10 K) difference spectrum shows the components visible only at T < 10 K. The shape of this difference spectrum exhibits a striking similarity to the EPR signal observed for horseradish peroxidase compound I, which has been assigned to a porphyrin radical spin coupled to a heme iron [34]. Recently, the same signal has been observed for the B-class DyP from *Klebsiella pneumoniae* (*Kp*DyP) at 2.5 K [23].

The (6–20 K) and (6–10 K) difference spectra show that species that are still detectable at 10 K but not at 20 K contribute to the EPR spectrum. The resonances resulting from these species are obtained by calculation of the (10–20 K) difference spectrum shown at the bottom of Figure 3B. This difference spectrum clearly shows the presence of two components. Firstly, an organic radical strongly broadened (width ~40 mT) by exchange coupling to the heme iron. The width and the temperature dependence of this signal resemble the exchange-coupled intermediate [(Fe^IV^=O Trp_321_^•+^] formed in *M. tuberculosis* KatG upon reaction with H_2_O_2_ or peroxyacetic acid [35], and a similar species formed in cytochrome c oxidase [36]. Consequently, we attribute this signal to an exchange-coupled tryptophan radical (Trp^•^) formed in *Dictyostelium* DyPA upon reaction with H_2_O_2_. Secondly, a narrower signal at *g* = 2.001 is observed, that exhibits a shoulder at lower fields, closely resembling EPR signals that have been reported to arise from protein-bound tyrosyl radicals (Tyr^•^) formed in heme peroxidases, see, e.g., [37,38,39,40,41]. Thus, we conclude that *Dictyostelium* DyPA uses both Trp as well as a Tyr radical chemistry in compound I formation.

The organic radical spectra recorded at 40 K (Figure 3C) resemble those observed for *Synechocystis* PCC6803 catalase-peroxidase obtained at 60 K under nonsaturating conditions [30]. However, *Dictyostelium* DyPA exhibits an additional spectral feature at *g* = 2.026, and the *g*^eff^ = 2.005 spectrum appears to exhibit a significantly larger overall width ~10 mT vs. 7.5 mT accompanied by a smaller peak-to-trough width of 14 mT vs. 19 mT observed for catalase-peroxidase. Nevertheless, this organic radical signal also did not show temperature-dependent changes in the spectral width [30] as observed here. Based on perdeuteration studies, contributions from both Trp^•^ and Tyr^•^ were identified for catalase-peroxidase [30]. Moreover, the *g*^eff^ = 2.005 signal closely resembles that observed in cytochrome c oxidase caused by Tyr^•^ [38]. Consequently, the free radical signals observed for *Dictyostelium* DyPA at 40 K support the presence of both types of protein-based radicals.

### 2.5. Substrate Specificities of Dictyostelium DyPA

DyPs catalyze many industrially desirable reactions and have dye-decolorizing as well as a general peroxidase activity. We examined the *Dictyostelium* DyPA activity toward several prototypic hydrogen donors and aromatic substrates in the presence of H_2_O_2_. Similar to other members of the DyP family [6,10,12], *Dictyostelium* DyPA displays greater activity in the acidic pH range with optimal turnover at pH 4.0 for substrates such as ABTS, pyrogallol, and veratryl alcohol (Figure 4A,B and Appendix A). For the anthraquinone-based dye RB4, the optimal pH is 3.0 (Figure 4C). Further, the thermal stability of the *Dictyostelium* DyPA was assessed by measuring the enzyme activity after 5 min incubation at a specific temperature. The thermal stability of *Dictyostelium* DyPA is comparable to that of bacterial DyPs in the presence of ABTS (7.5 mM) as a substrate. The maximum catalytic activity occurs in the temperature range of 20–40 °C (Figure 4D).

In the presence of 1 mM H_2_O_2_, *Dictyostelium* DyPA shows different apparent substrate affinities with the highest substrate specificity for RB4, followed by ABTS and pyrogallol (Figure 4E–Hand Table 2). The *k*_cat_/*K*_m_ for the general peroxidase substrate ABTS is 2.19 × 10^4^ M^−1^s^−1^ which is almost 10-fold higher than reported for class A and B enzymes and around 300–900 fold lower than that of class C and D enzymes [14]. The *k*_cat_/*K*_m_ for RB4 is 1.3 × 10^5^ M^−1^s^−1^ which is in the high range of enzymatic activities reported for class B enzymes (*k*_cat_/*K*_m_ = 10^2^–10^5^ M^−1^s^−1^). The values reported for *Rhodococcus jostii* RHA1 enzymes DypA and DypB are 10- and 1000-fold lower [16]. *Dictyostelium* DyPA activity toward anthraquinone dyes is around 2-fold lower than the reported class C (DyP2; RB5) and around 100-fold lower than class D (*Aau*DyPI, RB5) enzymes [6,12]. *Dictyostelium* DyPA can also oxidize NADH (*k*_obs_ 0.057 ± 0.001 s^−1^) and NADPH (*k*_obs_ 0.041 ± 0.003 s^−1^) at pH 4.0 but fails to oxidize Mn^2+^, a typical substrate for manganese peroxidases such as *Rj*DypB and Amycolatopsis sp. 75iv2 DyP2 [10,12].

### 2.6. Oxidation of ß-Aryl Ether Lignin Model Substrate and Veratryl Alcohol

To assess the lignin oxidizing properties of *Dictyostelium* DyPA, we used the β-aryl ether lignin model substrate **1** (Figure 5A). Similar to lignin oxidizing enzymes such as lignin peroxidases, laccases, and bacterial DyPs, *Dictyostelium* DyPA can oxidize guaiacylglycerol-β-guaiacyl ether (GGBGE) in the presence of H_2_O_2_ at pH 4.0. The reaction was monitored by reverse HPLC and thin-layer chromatography. We followed the development and increase in a second peak with a retention time of 18.8 min by analyzing aliquots of the reaction mixture at different time points (Figure 5B). Consistent with the concept that radical recombination leads to the formation of a higher-molecular-weight species, further analysis of the second peak by ESI-MS showed a m/z of 661.2 (Figure 5C), which corresponds to the mass of the sodium adduct of the predicted product (2). Dimerization of the lignin model substrate by various other DyPs has been reported [43,44]. It was suggested that dimerization is achieved by C─C coupling of free phenolic units leading to the formation of biphenyl compounds [19,44,45]. Indeed, such a mechanism is compatible with the results of our NMR measurements (Figure 5D–F). While we could not observe complete turnover of the racemic lignin model substrate, stereoselectivity was not observed in optical activity measurements and by chiral HPLC. Furthermore, we checked the oxidation of another lignin peroxidase model substrate veratryl alcohol and found that *Dictyostelium* DyPA oxidizes VA at pH 4.0 (Appendix A) with similar activity as reported for other DyPs [45]. The K_m_ for VA is 166 ± 58 µM (Figure 5G and Table 2).

### 2.7. Structural Features of Dictyostelium DyPA:O_2_, Dictyostelium DyPA:CN^−^ Complex and Dictyostelium DyPA:O_2_:VA Complexes

The structure of *Dictyostelium* DyPA in complex with an activated form of oxygen was determined to 1.7 Å resolution. Data collection, model, and refinement statistics are summarized in Appendix A. In the asymmetric unit, two molecules of *Dictyostelium* DyPA were found. Similar to chlorite dismutase and other DyPs, each monomer of *Dictyostelium* DyPA has two domains. Each domain comprises a 4-stranded antiparallel β-sheet, which is flanked by α-helices in a ferredoxin-like fold (Figure 6A) [10,12,17,46,47,48,49]. The *Dictyostelium* DyPA structure is more similar to the Class B/P bacterial DyP structures than to Class D/V eukaryotic structures (Appendix A). Comparisons with bacterial DyPs and eukaryotic DyP structures show that the *Dictyostelium* DyPA structure more closely resembles class B enzymes such as *Escherichia coli* O157 YfeX (PDB: 5GT2), *Klebsiella pneumoniae Kp*DyP (PDB: 6FKS), Vibrio *cholerae Vc*DyP (PDB: 5DE0), *Bacteroides thetaiotaomicron* VPI-5482 *Bt*DyP (PDB: 2GVK) than to *Rhodococcus jostii* RHA1 *Rj*DypB (PDB: 3QNS), and *Shewanella oneidensis* TyrA (PDB: 2IIZ). Backbone RMSD values correspond to 1.1, 1.14, 1.28, 1.4, 1.61, and 1.62 Å, respectively [10,23,46,47,50,51]. Higher backbone RMSD values for the largest superimposable core of the proteins were obtained with the bacterial class A/C and eukaryotic class D structures. For example, *Dictyostelium* DyPA shows an RMSD of 2.19 Å for 274 aligned residues to class A bacterial enzyme EfeB (*Escherichia coli* O157, PDB: 3O72), RMSD of 2.19 Å for 233 aligned residues to class C enzyme DyP2 (*Amycolatopsis* sp. ATCC 39116, PDB: 4G2C) [12,52], whereas *Dictyostelium* DyPA shows an RMSD of 2.29 Å and 2.38 Å for fungal class D enzymes *Bjerkandera adusta Bad*Dyp (PDB: 2D3Q; 290 Cα aligned) and *Auricularia auricula-judae*
*Aau*DyPI (PDB: 4AU9; 261 Cα aligned) [17,48] (Appendix A). Compared with the bacterial and *Dictyostelium* DyPA structures, eukaryotic DyPs structures have larger loops near the heme-binding pocket. This results in a deeper active site in class C and D enzymes [12,17].

Both monomers forming the *Dictyostelium* DyPA structure are nearly identical with a core Cα-RMSD of 0.10 Å. The heme is bound to the C-terminal region of each monomer. The Fe (III) is hexa-coordinated and is in the plane of the porphyrin ring. The heme group has His222 on its proximal side, and the distal side is occupied by Asp149, Arg239, Ser241, Leu253, and Phe255 (Figure 6B and Appendix A). Residue His222 makes close contact with Asp281, which can serve as a proton donor or acceptor for the imidazole ring [53]. Electron density analysis revealed excess electron density on the Nδ atom of His222 imidazole, which exceeds the maximum level of electron density on the carboxyl group of Asp281 by ~1σ. This observation suggests that the imidazole ring of His222 is negatively charged in the crystal structure of the *Dictyostelium* DyPA:O_2_ complex, while the carboxyl group of Asp281 is protonated. Asp149 has been predicted to have a similar function as the distal glutamate of chloroperoxidase [54]. Asp149 and Arg239 are conserved in all known DyPs [15]. The sixth coordination position at the distal face of the heme iron is occupied by an activated oxygen molecule with elongated bond distances between the oxygen atoms of 1.7 and 2.4 Å in the two monomers. The observed distances between the coordinated oxygen atom and the heme iron in the two monomers are 2.3 and 2.6 Å. The other oxygen atom is coordinated by a hydrogen-bond network that involves Asp149, Ser241, an ethylene glycol, and water molecules. The negative charge on His222 explains the elongated bond length in the activated O_2_ molecule by the decreasing effect of an additional electron on the energy difference between the highest occupied (HOMO) and lowest unoccupied (LUMO) molecular orbitals of the complex. The LUMO represents an antibonding π* molecular orbital of the oxygen molecule. The smaller energy gap between HOMO and LUMO leads to the population of the antibonding π* MO of the coordinated oxygen molecule and its activation. This activation results in weakening of the O-O bond to the point of cleavage [53].

Arg239 forms hydrogen bonds with the distally positioned heme propionate, Asp149, and the O_2_ molecule (Figure 6B). Similar to *Vc*DyP, the third residue at the distal side of the *Dictyostelium* DyPA heme is Ser241 instead of Asn246 in the case of *Rj*DypB. This substitution provides a slightly larger space between and more flexibility in the orientation of Asp149 and Ser241. The *Dictyostelium* DyPA Asp149 side chain is slightly rotated and is closer to the heme-Fe(III) atom and can function as an acid-base catalyst (The distance between the Asp side chain and the iron atom is 4.74 Å for *Vc*DyP, 5.05 for *Rj*DypB, and 4.7 Å for *Dictyostelium* DyPA) (Appendix A). The importance of the distal aspartate and arginine residues for catalysis has been studied by mutagenesis and structural approaches in other DyPs [26,55]. It was proposed that the role of catalytic aspartate differs in different classes of DyPs. Mutational studies show that the aspartate is essential for the formation of compound I in the class D enzyme *Bad*DyP [26], the conserved distal arginine is essential for peroxidase activity in the class B enzyme *Rj*DypB [55], and both distal aspartate and arginine are essential in the class B enzyme *Vc*DyP [51]. These findings suggest that there is functional diversity within the same class despite close structural resemblance.

To understand the binding mode of H_2_O_2_, we crystallized and solved the structure of *Dictyostelium* DyPA-CN^−^ to 1.85 Å. The asymmetric unit contains two copies of the complex. Both copies share nearly the same overall architecture, as indicated by RMSD of 0.128 Å (Appendix A–C). Though the binding mode of H_2_O_2_ and cyanide differ, the position of the carbon atom still mimics the position of the iron-coordinated oxygen of H_2_O_2_ during the formation of compound I [25]. Therefore, the position of the cyanide can provide information about possible interactions between Asp149 and the iron-coordinated oxygen of H_2_O_2_. Superposition of the *Dictyostelium* DyPA:O_2_ and the cyanide complex structures shows a minor change in Cα-RMSD of 0.13 Å (Appendix A). The cyanide molecule takes the place of the O_2_ molecule in the DyPA:O_2_ complex structure. No changes in the conformation of active site residues are observed (Figure 6C and Appendix A). This is different from the situation reported for the *Bad*DyP-CN^−^ complex, where a change in the location of the aspartate side chain was reported [26]. This led to the proposal that the swinging of the aspartate residue is required for the compound I formation and that completion of reaction requires the aspartate to move back to its initial position. In agreement with this concept, two distinct conformations of the aspartate side chain were observed in the native structure of *Aau*DyPI [26,48]. Our O_2_ complex structure, as well as the CN^−^ complex structure, shows the Asp149 in hydrogen bonding distance to the proximal oxygen of H_2_O_2_, suggesting that Asp149 can accept a proton from H_2_O_2_ and compound I can form without side-chain movement. Using serial femtosecond X-ray crystallography, Lucic et al. have determined DtpB structures in resting (Fe^III^) as well as in compound I state (Fe^IV^=O and a porphyrin cation radical). Moreover, using mutagenesis experiments, they went on to show a catalytic role for the distal arginine residue in the formation of compound I [24]. However, compared to the DtpB resting state structure, Fe^III^ is hexa-coordinated in our *Dictyostelium* DyPA:O_2_ complex structure and has an elongated O_2_ molecule at the distal face of heme. In the case of *Dictyostelium* DyPA, both the aspartate and arginine residue on the distal face are in an optimal position (Figure 6B,C) to take on a catalytic role during compound I formation upon H_2_O_2_ addition. Since discrepancies still exist regarding the mechanistic roles of the distal aspartate and arginine residues during compound I formation, further mutagenesis and biochemical experiments are required to assign to these residues a definite role in the formation of *Dictyostelium* DyPA compound I.

Since *Dictyostelium* DyPA exists as a dimer in solution as well as in the crystal structure, we analyzed the dimer interface using the ‘Protein interfaces, surfaces and assemblies’ (PISA) service at the European Bioinformatics Institute website [56]. The dimer interface has an inaccessible area of approximately 1274 Å^2^ for chain A and chain B (9% of each subunit surface). A head-to-tail interaction was observed between the monomers. The dimer interface can be further divided into two identical subinterfaces and interactions at one subinterface are described here (Figure 6D,E). The main interaction between monomers is mediated by helix α5 of chain A, which is in contact with β1, β4, with the loop between α5/β4 and α7 of chain B. Chain A, β4 interacts with the loop between β6/β7 of chain B. Direct interactions involving several hydrophobic, hydrogen bond interactions, and ionic interactions are shown in Figure 6E. Besides these interactions, there are many indirect hydrogen bond interactions between the chains involving water molecules (Appendix A).

Next, to identify potential substrate-binding pockets, we used the program POCASA [57]. Out of several suggested binding pockets, six binding pockets close to the heme are shown in Figure 7A. The heme can be directly accessed from pockets 1 and 2, which are smaller in size. Pocket 1 is lined by two channels that can create a link between the heme and the surface of the enzyme. The second channel is branching out from the first one, and both channels are ~21–24 Å away from the heme (Figure 7B,C). On the enzyme surface, the first channel entrance is formed in part by residues Arg56 and Pro8, while the second channel entrance is formed by Met9, His10, Glu141, Gly150, Asn153, and Gln240 (Figure 7B,C). These solvent channels are lined by charged and polar residues. Pocket 2 is near to the third shallow propionate channel, which includes both propionate moieties of the heme group and a water molecule and is lined by the Asp227, Lys236, Glu152, Arg204, and Glu211 (Figure 7D). This shallow propionate pocket is highly conserved across DyP classes, whereas the distal channels are quite diverse. Distal channels are present in classes B, C, and D but absent in class A. While the distal channels are too narrow to accommodate bulkier substrates, H_2_O_2_ can reach the heme cofactor via these channels to activate the enzyme by compound I formation. Pockets 4, 5, and 6 located at the dimer interface can accommodate a range of larger substrate molecules (Figure 7A). As these sites are not in direct contact with the heme group, long-range energy transfer (LRET) from the surface-bound substrate to the heme is required for enzymatic turnover.

Typically, the LRET involves a surface-exposed tryptophan [58] or tyrosine residue. The latter was reported for *Trametopsis cervina* LiP [39]. *Dictyostelium* DyPA has ten tyrosine and two tryptophan residues. Out of these, seven tyrosine and both tryptophan residues are surface exposed (Figure 7E). To serve as an LRET-mediated oxidation site on the enzyme surface, the distance of the exposed residue from the heme molecule needs to be short, and electron transfer should be facilitated via stacked aromatic residues. Furthermore, cation radical formation is facilitated by the presence of a negatively charged amino acid in close proximity of the surface-exposed aromatic residue [59,60]. According to these criteria, Tyr90, Tyr191 Tyr244, Tyr286, and Trp190 could be involved in LRET which is further supported by EPR data suggesting that the *Dictyostelium* DyPA compound I involves the formation of both a Trp^•^ as well as a Tyr^•^ radicals, which may contribute to an enhanced reactivity toward recalcitrant substrates that require oxidation potentials that cannot be realized at the heme site.

In the case of *Aau*DyPI, a pathway connecting the heme group and surface residue Trp337 was shown to be essential for oxidation of Reactive Blue 19 [61], while a second surface exposed substrate binding site that comprises residue Tyr244 was proposed to involve LRET to the heme group [48,62]. Tyr244 and other residues contributing to this pathway are well-conserved between *Aau*DyPI and *Dictyostelium* DyPA (Figure 7E, inset 1). Potential LRET pathways for *Dictyostelium* DyPA from surface-exposed residues to heme are shown in Figure 7E (Inset 1 and 2).

To gain insight into the exact location of substrate binding sites, we solved the structure of the *Dictyostelium* DyPA:O_2_:VA complex structure to 1.6 Å resolution. The asymmetric unit contains two copies of the complex, having the same overall architecture (Cα-RMSD of 0.105 Å) (Appendix A). The structure of the *Dictyostelium* DyPA:O_2_:VA complex is nearly identical to the *Dictyostelium* DyPA:O_2_ structure (Cα-RMSD 0.15 Å). Each monomer has two bound veratryl alcohol molecules, which are located close to the POCASA predicted substrate-binding pockets 4 and 6 at the dimer interface. Their distance from the iron atom of the heme group corresponds to approximately 22 Å (Figure 8D). The VA binding sites are quite different from those published for the ascorbic acid (ASC) and 2, 6-dimethoxyphenol (DMP) binding site of *Bad*DyP, which occupies a shallow pocket near the γ-edge of the heme (Figure 7A,B) [63]. In the case of *Aau*DyPI, two imidazole (IMD) binding sites were reported [64]. The position of the first imidazole overlaps with the space required for binding of H_2_O_2_ in the heme cavity, and the second imidazole sits in a cavity close to the entrance of the heme-access channel (Figure 8C).

In our *Dictyostelium* DyPA:O_2_:VA complex structure, two binding pockets are well defined. VA binding pocket 1 (POCASA^pocket6^) is made up of residues Lys188^A^, Tyr191^A^, Ile247^A^, Thr248^A^, Gln116^B^, Met119^B^, Glu124^B^, and Ile129^B^. VA binding pocket 2 (POCASA^pocket 4^) is composed of residues Val139^A^, Glu140^A^, Ile48^B^, Ile51^B^, Thr110^B^, and Lys113^B^. Minor rearrangements of the side chains are required for VA binding (Figure 8D, inset). Pocket 1, pocket 2, and the pathways leading to the heme center are nearly identical for chain B (Appendix A). The veratryl alcohol molecule 1 (VA1) has full occupancy and is hydrogen bonded to Tyr191^A^ via a water molecule. Electron transfer from VA1 to the heme moiety can occur via a hydrogen-bonding network, as shown in Figure 8E. Alternatively, LRET with the involvement of the aromatic amino acids Tyr191, Trp190, and Phe147 is possible. It is interesting to note that the Tyr191 is not conserved within the DyP family. This suggests that aromatic residues contributing to LRET do not necessarily have to be in the exact same position. A similar oxidation pathway has been reported for *Amycolatopsis* sp. *75iv2* DyP2, where a tyrosine residue is present near the Mn^2+^ binding site that appears to facilitate electron transfer between heme and Mn^2+^ [12]. In the case of *Aau*DyPI, a surface tyrosine and tryptophan-based radical center were reported [63]. Our result is in line with the previously reported LRET pathways in lignin peroxidase (LiP) and versatile peroxidases (VP) [65,66,67]. Using QM/MM approaches and mutagenesis experiments, the authors reported a possible LRET pathway from the substrate to the heme center involving electron transfer via 3 aromatic amino acids [66]. For example, in the case of VP, surface Trp164, neighboring Phe198, and buried Trp244 are involved in the LRET pathway, whereas Trp171^surface^, Phe205^neighboring^, and Trp251^burried^ constitute the LRET pathway in LiP. The distance from the protein surface to the heme center of VP is around ~15 Å, which matches our observations [66]. However, no structural data are available for VP and LiP that provide information about their substrate–binding pockets. From the second VA molecule, the distal side of heme can be reached via a hydrogen-bonding network involving several water molecules and residues Glu141, Arg137, and Asp149 (Figure 8F). Hydrogen-bonding networks have been reported to serve as proton transfer pathways in the case of *Bad*DyP:DMP/ascorbic acid complexes and ascorbate peroxidase–ascorbic acid complex [68]. However, further structural and functional studies are required to validate these LRET pathways.

## 3. Materials and Methods

### 3.1. Chemicals

1-amino-4-[3-(4,6-dichlorotriazin-2-ylamino)-4-sulfophenylamino]anthraquinone-2-sulfonic acid (Reactive blue 4, RB4), 1,2,3-trihydroxybenzene (Pyrogallol), 2, 2′-Azino-bis (3-ehtylbezothiazoline-6-sulfonic acid (ABTS), 3,4-Dimethoxybenzyl alcohol (Veratryl alcohol), H_2_O_2_, β-Nicotinamide adenine dinucleotide, reduced disodium salt hydrate (NADH), and β-Nicotinamide adenine dinucleotide 2′-phosphate reduced tetrasodium salt hydrate (NADPH) were purchased from Sigma-Aldrich, and 1-(4-hydroxy-3-methoxyphenyl)-2- (2-methoxyphenoxy)propane-1,3-diol (Guaiacylglycerol-β-guaiacyl ether; GGBGE) was purchased from Tokyo Chemical Industry UK Ltd.

### 3.2. Protein Purification

*Dictyostelium* DyPA was expressed and purified from *Escherichia coli* Rosetta(DE3)pLysS cells as described previously [20]. Protein concentrations were determined using the Bradford protein assay with bovine serum albumin (BSA) as a standard. Concentrations relate to the monomers throughout the text. Heme concentration was determined by the pyridine hemochromogen method.

### 3.3. Plasmid Construction, Cell Culture, and Fluorescence Microscopy

*Dictyostelium* DyPA expression constructs with N- and C-terminal EYFP fusions were generated in the plasmids pDXAYFPmcs and pDXAmcsYFP, respectively [69]. The DNA fragment encoding DyPA cDNA was inserted between BamHI and XhoI sites by conventional PCR using *Dictyostelium* gDNA. All the constructs were verified through DNA sequencing.

*Dictyostelium discoideum* AX2 cells were grown in HL-5C medium (Formedium) at 21°C. Cells were transformed with the expression constructs by electroporation as described previously [70,71]. Transformants were selected in the presence of 10 µg/mL G-418 (Formedium). *Dictyostelium discoideum* AX2 cells were grown on glass-bottom petri plates (MatTek Corp) to 50–60% confluency for confocal microscopy imaging. Imaging was performed in a buffer containing 10 mM MES-NaOH pH 6.5, 2 mM MgCl_2_, 0.2 mM CaCl_2_, at 512 nm with a Leica TCS SP2 confocal laser scanning microscope equipped with a 63 × 1.4 NA HCX PL APO CS oil immersion objective. Experiments were performed at room temperature.

### 3.4. UV-Visible Spectroscopy

A Cary 50 or Shimadzu UV-2400 spectrophotometer was used to record absorption spectra of *Dictyostelium* DyPA at 25 °C with a spectral bandwidth of 1.0 nm using 1 cm pathlength cuvette. To study the formation of compound I or to analyze the effect of cyanide on *Dictyostelium* DyPA, 10 µM *Dictyostelium* DyPA was mixed with 10 µM of H_2_O_2_ or 5 mM KCN in 50 mM Tris-HCl pH 8.0 and 150 mM NaCl at 25 °C. For pH-dependent measurements, we performed assay in the buffer solutions containing 150 mM NaCl and either 50 mM sodium citrate (pH 3.0), 50 mM sodium acetate (pH 4–5.0), 50 mM Bis-Tris (pH 6.0), 50 mM Hepes (pH 7.0), or 50 mM Tris-HCl (pH 8.0–9.0).

### 3.5. Analytical Ultracentrifugation

Sedimentation velocity experiments were carried out in a Beckman Coulter ProteomeLab XL-I analytical ultracentrifuge at 50,000 rpm and 20 °C, in a buffer containing 50 mM Tris-HCl pH 8.0 and 150 mM NaCl, using an An-50 Ti rotor. Protein concentration profiles were measured using UV absorption scanning optics at 280 nm and the data acquisition software ProteomeLab XL-I GUI 6.0 (firmware 5.7, Beckman Coulter Life Sciences, Indianapolis, IN, USA). Experiments were performed in 3 or 12 mm double sector centerpieces filled with 100 µL or 400 µL samples, respectively. Data were analyzed using a model for diffusion-deconvoluted differential sedimentation coefficient distributions [c(s) distributions] implemented in SEDFIT [21]. Partial specific volume, buffer density, and viscosity were calculated by the program SEDNTERP [72] and were used to correct the experimental sedimentation coefficients to s_20,w_. Sedimentation coefficient distributions were converted to 12 mm path length for better comparison. Contributions of bound heme to the partial specific volume of *Dictyostelium* DyPA were not taken into account.

### 3.6. Stopped-Flow Kinetics

Transient kinetic experiments were performed at 25 °C with a Hi-tech Scientific SF-61 DX stopped-flow system (TgK Scientific Limited, Bradford-on-Avon, U.K.). In total, 10 µM of the enzyme was mixed with an equal volume of H_2_O_2_ at various H_2_O_2_ concentrations. The rate of decay of the Soret band upon H_2_O_2_ addition was monitored at wavelength 400 nm, and the data were fitted to obtain a pseudo-first-order rate constant (*k*_obs_). The second-order rate constant for the formation of compound I was evaluated from plots of *k*_obs_ versus H_2_O_2_ concentration. All reactions were performed in a buffer containing 50 mM Tris-HCl pH 8.0 and 150 mM NaCl at 25 °C. All measurements were performed at least in triplicate.

### 3.7. Electron Paramagnetic Resonance Spectroscopy

EPR spectra at 9.4 GHz (X band) were recorded on a Bruker ELEXSYS E580 spectrometer equipped with Super High Sensitivity Probe Head (V2.0). Temperature control was achieved with a continuous flow liquid helium cryostat (Oxford Instruments ESR900) controlled by an Oxford Intelligent Temperature Controller ITC 503S. The EPR spectra were recorded from samples containing ~250 µM *Dictyostelium* DyPA in solution. In total, 50 µL were filled into 3 mm diameter EPR tubes and frozen in liquid nitrogen prior to the experiments. Unless otherwise stated, the parameters for the EPR experiments were as follows: microwave frequency = 9.40 GHz, modulation amplitude = 0.5 mT, modulation frequency = 100 kHz, temperature = 6 K, and microwave power = 1 mW. The EPR spectra were obtained as an average of 5–10 scans with a sweep time of 168 s with a time constant of 20.48 ms (8192 data points). The scan range was 50–450 mT.

### 3.8. EPR Spectra Simulation

Simulation of the *Dictyostelium* DyPA EPR spectrum recorded at 6 K was carried out using the function “pepper” of the software package EasySpin (version 4.5.0, see https://easyspin.org/forum/ for further details, accessed on 1 December 2020), developed by Stoll and Schweiger [73]. Line widths have been accounted for only by broadening due to unresolved hyperfine couplings, specified in the orientation-dependent parameter (tensor) *HStrain* in pepper. For calculation of the power spectra, 60 orientations have been included (option *nKnots*), corresponding to 1.5-degree increments. For details of the underlying algorithms, see [73].

### 3.9. Steady-State Kinetic Measurements

Steady-state kinetic measurements were performed spectrophotometrically using a SPECTROstar Omega plate reader (BMG Labtech GmbH, Ortenberg, Germany). The standard assay was executed in 100 µL of 50 mM sodium acetate pH 4.0 and 150 mM NaCl at 25 °C, containing 10 mM ABTS, 1.0 mM H_2_O_2_ and with the appropriate amount of protein. The reaction was initiated upon the addition of 1 mM H_2_O_2_ and was monitored at 414 nm (ε_414_ = 36.6 mM^−1^ cm^−1^). Steady-state kinetic parameters were determined for ABTS (ε_414_ = 36.6 mM^−1^ cm^−1^), pyrogallol (ε_430_ = 2.47 mM^−1^ cm^−1^), reactive blue 4 (RB4, ε_610_ = 4.2 mM^−1^ cm^−1^), and veratryl alcohol (ε_310_ = 9.3 mM^−1^ cm^−1^). RB4 assay was performed in a buffer containing 50 mM sodium citrate pH 3.0 and 150 mM NaCl. Kinetic parameters were obtained by fitting the data to the Michaelis–Menten equation using OriginPro 9.6. All assays were performed at least in triplicate. For pH optimization measurements, we used buffer solutions containing 150 mM NaCl and either 50 mM sodium citrate (pH 3.0–3.2), 50 mM sodium acetate (pH 4–5.0), 50 mM Bis-Tris (pH 6.0), 50 mM Hepes (pH 7.0), or 50 mM Tris-HCl (pH 8.0–9.0).

### 3.10. Thermal Stability of Dictyostelium DyPA

*Dictyostelium* DyPA was incubated in 50 mM potassium phosphate buffer pH 7.5, containing 150 mM NaCl for 5 min at temperatures in the range from 30–90 °C and slowly brought back to room temperature. To check for residual enzyme activity, steady-state kinetic assays were performed at pH 4.0 and 25 °C as described above with 7.5 mM ABTS as a substrate.

### 3.11. Oxidation of β-aryl Ether Lignin Model Substrate

The model lignin substrate (Guaiacylglycerol-β-guaiacyl ether; Tokyo Chemical Industry UK Ltd., Oxford, U.K.) was prepared, as described previously [44], in a buffer containing 50 mM sodium acetate pH 4.0 and 150 mM NaCl, 10 µM of the enzyme was added to the lignin model substrate, and 2 mM H_2_O_2_ was used to start the reaction. Following incubation for 3 h at room temperature, samples were heated at 95 °C for 5 min and then centrifuged at 13,000 rpm for 15 min to remove the precipitated enzyme. The supernatant was loaded on a C18 reverse-phase HPLC column (Prontosil 120-5 C18, 5 µm, 250 × 4.6 mm) at a flow rate of 1 mL/min, and the elution profile was monitored at 254 nm. A linear gradient of 30 to 90% of methanol in water was used over 30 min, and the second peak corresponding to the product was analyzed by ESI-MS(+).

For NMR studies, 5 mg of the substrate was dissolved in 200 µL of acetone, added to 2.8 mL of buffer containing 50 mM sodium acetate pH 4.6 and 150 mM NaCl. *Dictyostelium* DyPA was added to the lignin model substrate up to a final concentration of 10 µM, and subsequently, 1 mM hydrogen peroxide was added 5 times with an interval of 15 min. The reaction was performed at room temperature and monitored by thin-layer chromatography. The reaction was stopped by adding an equal volume of dichloromethane (DCM), and the enzyme was removed by centrifugation at 10,000× *g* for 5 min. The reaction mixture was evaporated, dissolved in little DCM, and filtered before it was subjected to flash column chromatography (FC). Preparative FC was performed on a MPLC-Reveleris system from Büchi using a 4g-silica cartridge. Eluent system: DCM/MeOH gradient. Fractions were analyzed by TLC and LCMS to identify dimer-containing fractions, which were evaporated to obtain the purified solid reaction product.

Dimer methylation: The obtained dimer (10 mg, 16 µmol) was dissolved in 1 mL acetone, and subsequently, 1 mg K_2_CO_3_ and 75 µmol methyl iodide were added. The reaction mixture was stirred at room temperature for 20 h. The solvent was evaporated, and the residual solid was redissolved in DMSO-d6 for NMR analysis.

### 3.12. NMR Spectroscopy

Nuclear magnetic resonance (NMR) spectra were recorded at room temperature if not stated otherwise. Spectra were recorded on either a Bruker Ascend 600 MHz with an Avance NEO Console, Sample Case, and Cryo-Probe DUL or a Bruker Ultrashield 500 MHz with Avance IIIHD Console, Sample Xpress, and Cryo-Probe TCI or a Bruker Ascend 400 MHz with Avance III Console, Sample Xpress, and Prodigy BBFO probe. Chemical shifts are reported relative to solvent signal (DMSO-d6: δH = 2.50 ppm, δC = 39.52 ppm). Signals were assigned by first-order analysis, and assignments were supported by two-dimensional 1H, 1H and 1H, 13C correlation spectroscopy (COSY, HSQC, HMBC, and NOESY).

**^1^H NMR**: 7.05 (dd, 1H, ^3^J_HH_ = 4.3 Hz, ^4^J_HH_ = 1.7 Hz, A2-H), 7.01 (dd, 1H, ^3^J_HH_ = 4.6 Hz, ^4^J_HH_ = 1.7 Hz, A’2,H), 6.98 (dd, 2H, ^3^J_HH_ = 6.8 Hz, ^4^J_HH_ = 1.5 Hz, B5-H, B’5-H), 6.90–6.88 (m, 2H, B2-H, B’2-H), 6.85–6.77 (m, 4H, B3-H, B’3-H, B4-H, B’4-H), 6.73 (t, 1H, ^4^J_HH_ = 1.8 Hz, A6-H), 6.66 (t, 1H, ^4^J_HH_ = 1.9 Hz, A’6-H), 4.76 (d, 1H, ^3^J_HH_ = 5.2 Hz, 9-H), 4.74 (d, 1H, ^3^J_HH_ = 5.2 Hz, 9′-H), 4.35–4.32 (m, 2H, 7-H, 7′-H), 3.80 (s, 3H, A’3-OCH_3_), 3.78 (s, 3H, A3-OCH_3_), 3.66 (m, 6H, B3-OCH_3_, B’3-OCH_3_), 3.62 (m, 4H, 8-H_2_, 8′-H_2_) 3.45 (m, 2H, A4-OCH_3_); **^13^C-NMR** 151.7 (A3),149.8 (B1/’1), 149.7 (B1/’1), 148.1 (B’6), 148.0 (B6), 146.8 (A’3), 145.4 (A4), 142.7 (A’4), 137.1 (A1), 132.1 (A’1), 124.9 (A5), 121.8 (A6), 121.5 (A’6), 121.1 (B4/B’4), 120.9 (B4/B’4), 120.6 (A’5), 116.1 (B5/B’5), 115.8 (B5/B’5), 112.6 (B2, B’2), 110.8 (A2), 109.8 (A’2), 83.8 (7′), 83.6 (7), 71.6 (9, 9′), 60.2 (8′), 60.0 (8), 59.7 (A4-OCH_3_), 55.7 (A’3-OCH_3_), 55.5 (B1-/B’1-/A3-OCH_3_), 54.9.

### 3.13. LC-MS Analysis

Analytical reverse-phase HPLC (MeCN/water, 0.05% TFA) was performed on an Acquity H UPLC system (Waters, Milford, USA) with an Acquity UPLC BEH C18-column (2.1 x 50 mm, Waters, Milford, USA). Molecular masses and purity were confirmed by electrospray mass spectrometry using an Acquity QDa (Waters, Milford, USA) detector in positive ionization mode.

### 3.14. Crystallization, Data Collection, and Structure Determination

*Dictyostelium* DyPA crystals were grown at 20 °C, using vapor diffusion in a hanging drop setup, as described previously [20]. In total, 2 µL DyPA (10 mg/mL) was mixed with 2 µL of reservoir solution, and after one week, crystals appeared in 2.4 M sodium malonate pH 7.0. The crystals were briefly soaked in a reservoir solution supplemented with 15% ethylene glycol and then flash-frozen directly in liquid nitrogen. Data were collected at the European Synchrotron Radiation Facility (ESRF, Grenoble) on beamline ID29. Crystals grew in the space group *P*4_1_2_1_2. Data were indexed, processed, and scaled with XDS [74]. The structure of *Dictyostelium* DyPA:O_2_ complex was determined by molecular replacement using Phaser [75]. *Shewanella oneidensis* TyrA (PDB: 2IIZ) structure was used as a search model [46]. An initial model was built with Coot [76] and refined in REFMAC5 [77] from the CCP4 program suite [78] or with phenix.refine using refined Translation/Libration/Screw tensors [79]. Further improvements were achieved by successive cycles of model building and refinement.

CN^−^-complexed/O_2_:VA-complexed crystals were prepared by adding 5 mM KCN/50 mM VA to the protein (10 mg/mL). They were grown in 2.4 M sodium malonate pH 7.0 in a hanging drop setup at 20 °C. These complex crystals diffracted to 1.85 Å and 1.6 Å, respectively. For the complex structure determination, the *Dictyostelium* DyPA native structure was used as a starting model. Data collection, processing, and refinement statistics are summarized in Appendix A. Structural figures were generated in PyMOL (The PyMOL Molecular Graphics System, Version 2.4.1, Schrödinger Inc., New York, NY, U.S.A.).

### 3.15. Bioinformatics

Multiple sequence alignments were generated using Clustal Omega [80]. Protein interaction interfaces were examined using the PDBePISA server (Proteins, Interfaces, Structures, and Assemblies; PISA) [56]. The POCASA webserver was used for the examination of substrate binding pockets [57]. The DALI server was used for structural comparison [81].

## 4. Conclusions

We describe the comprehensive biochemical and structural characterization of a cytosolic dye-decolorizing peroxidase from *Dictyostelium discoideum*. *Dictyostelium* DyPA is a dimer, with each monomer exhibiting a two-domain, α/β ferredoxin-like fold. The enzyme shows greater structural similarity to the “primitive” class P(B) DyP superfamily members produced by bacteria than to the “advanced” fungal DyPs of class V(C,D). UV-Vis and EPR spectroscopy identified the presence of a high-spin iron-containing heme that forms a protein-based radical upon H_2_O_2_ addition. *Dictyostelium* DyPA uses both Trp as well as a Tyr radical chemistry in the catalytic processing of its substrates. Lignin oxidation, dye decolorization, and general peroxidase activity were observed for *Dictyostelium* DyPA. The crystal structures of *Dictyostelium* DyPA in complex with either O_2_ or CN^−^ show that Asp149 is in an optimal position to accept a proton from H_2_O_2_ during the formation of compound I. Moreover, we report a DyP structure with the lignin model compound veratryl alcohol and delineate a plausible LRET pathway from the substrate binding site to the heme center, which can now be validated by combining mutagenic and time-resolved spectroscopic studies.

## Figures and Tables

**Figure 1 ijms-22-06265-f001:**
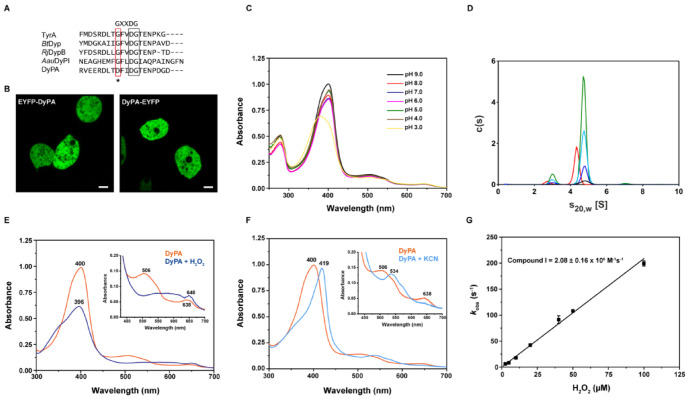
Cellular localization and biochemical properties of *Dictyostelium* DyPA. (**A**) Multiple sequence alignment of conserved GXXDG motif of different DyP-type peroxidases. Conserved glycine of the GXXDG motif is shown in a red box and DG in a black box. The asterisk (*) marks the position where the substitution of the first conserved glycine residue by aspartate (D146) occurs in *Dictyostelium* DyPA. (**B**) Confocal images of live *Dictyostelium discoideum* cells show the cytoplasmic localization of fusion constructs EYFP-DyPA and DyPA-EYFP; scale bars 5 µm. (**C**) Electronic absorption spectra of *Dictyostelium* DyPA at different pH. (**D**) Characterization of the oligomerization state of *Dictyostelium* DyPA by analytical ultracentrifugation. Sedimentation velocity runs at 50,000 rpm and 20 °C were performed with the following concentrations *of Dictyostelium* DyPA 2.1 µM (black), 6.3 µM (blue), 18.9 µM (light blue), and 33.6 µM (green) and 20 µM apo- *Dictyostelium* DyPA lacking the heme cofactor (red) respectively. Independent of the protein concentration, the main fraction of *Dictyostelium* DyPA sediments as a dimer with s_20,w_ = 4.8 S. At low heme saturation, the sedimentation coefficient decreases to 4.3 S, indicating the formation of *Dictyostelium* DyPA dimers with a less compact shape. (**E**) The electronic absorption spectrum of Fe^III^-*Dictyostelium* DyPA and compound I. Spectra of 10 µM *Dictyostelium* DyPA in the absence (orange) and presence (blue) of 10 µM H_2_O_2_. The inset displays Q and CT bands. (**F**) Electronic absorption spectra of 10 µM *Dictyostelium* DyPA in the absence (orange) and presence (light blue) of 5 mM KCN. Q and CT-bands are shown in the inset. (**G**) Stopped-flow analysis of the association of H_2_O_2_ with the *Dictyostelium* DyPA. The slope of the observed rate constants (*k*_obs_) plotted against the H_2_O_2_ concentration defines the second-order rate constant for compound I formation as 2.08 ± 0.16 × 10^6^ M^−1^s^−1^.

**Figure 2 ijms-22-06265-f002:**
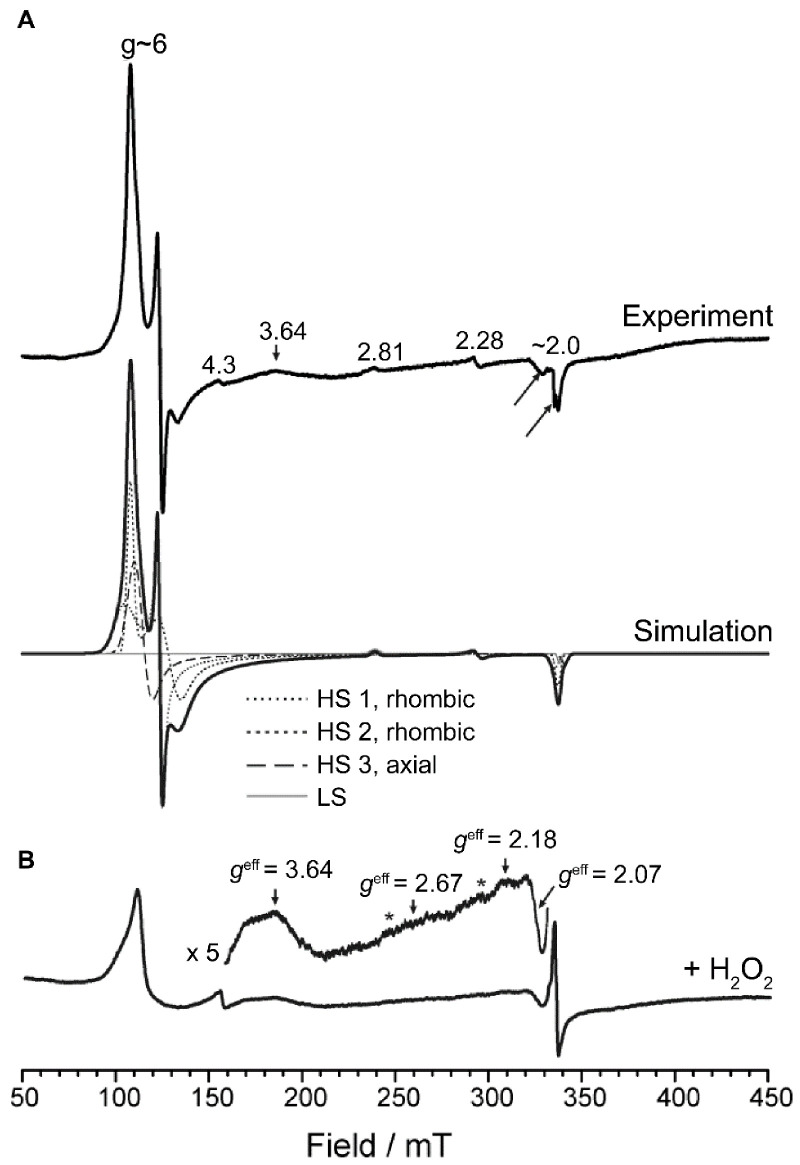
EPR spectrum of *Dictyostelium* DyPA. (**A**) EPR spectrum of *Dictyostelium* DyPA (top) and simulated spectrum (bottom, black, and solid). The single-component spectra of the simulation are shown as dotted lines (HS 1), short dashed lines (HS 2), dashed lines (HS 3), and in grey (LS). The signal at *g* = 4.3 is caused by a small amount of adventitious iron. Two features in the *g* = 2 (arrows) region are due to a cavity contaminant. A baseline distortion evident mainly in the region 150–400 mT originates from a small amount of solid air in the sample tube. (**B**) *Dictyostelium* DyPA incubated for 5 s with 3 mM H_2_O_2_. The experimental conditions were equal to those used for obtaining the spectrum in panel A. Spurious amounts of LS left after H_2_O_2_ treatment are marked by stars in the inset. The experimental spectra in panels A and B are drawn to scale.

**Figure 3 ijms-22-06265-f003:**
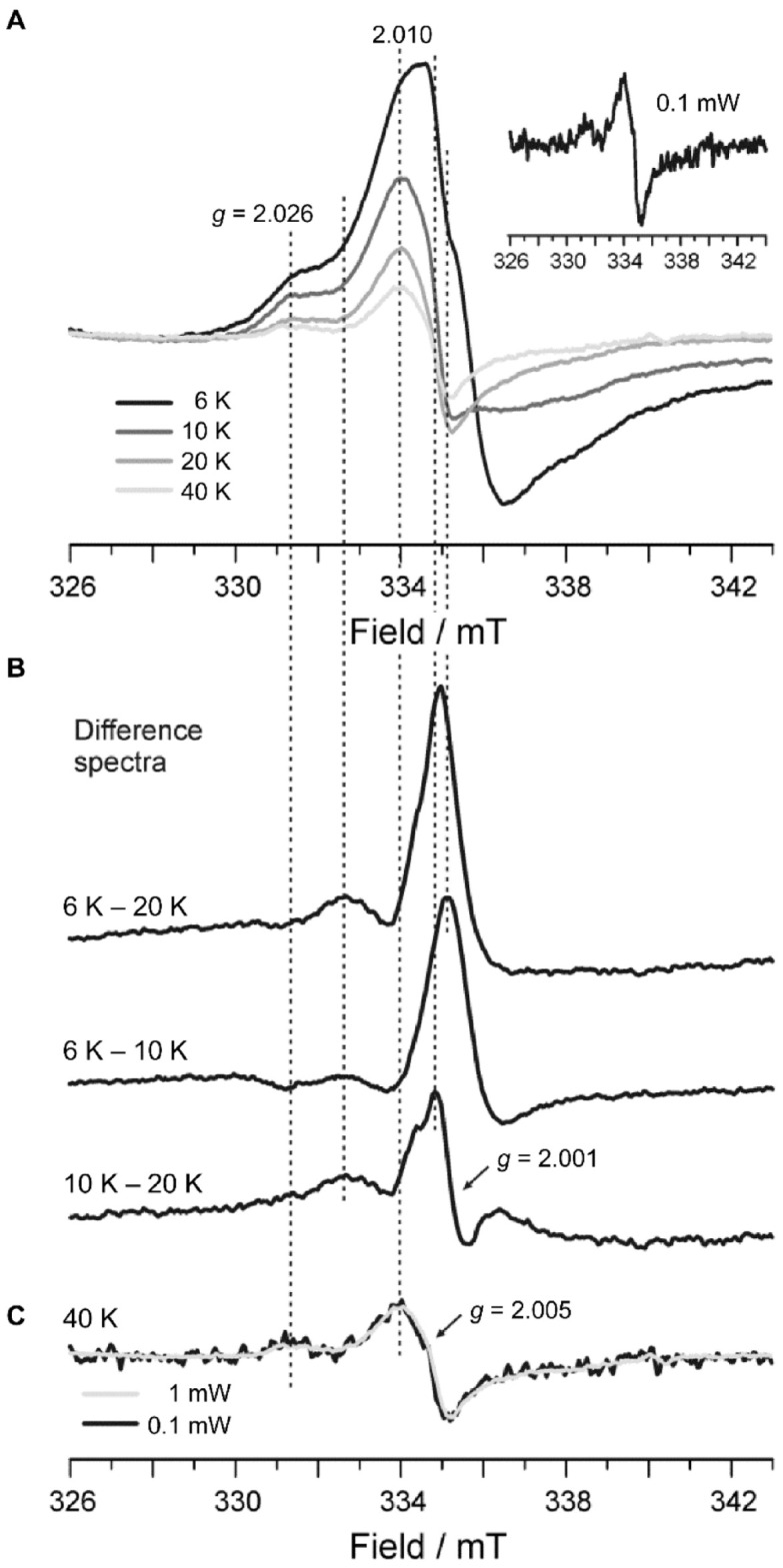
EPR analysis of organic radical formation in *Dictyostelium* DyPA after addition of H_2_O_2_. (**A**) Temperature dependence of the g ≈ 2.00 signal originating from organic radicals observed in *Dictyostelium* DyPA after reaction with H_2_O_2_. The right inset shows the spectrum recorded at 40 K under nonsaturating conditions (microwave power = 0.1 mW, modulation amplitude = 0.1 mT, 15 averages) from the same sample. (**B**) Difference spectra from the temperature dependence data shown in panel A, corrected for temperature effects on the signal amplitude (due to temperature effects on the Boltzmann distribution of spin states). The (6–20 K) difference spectrum contains all components stable at T < 20 K, and the (6–10 K) difference spectrum the components stable only at T < 10 K. The (10–20 K) difference spectrum shown at the bottom consequently displays those components still present at 10 K but absent at 20 K. (**C**) Overlay of the 40 K spectra recorded with microwave powers of 1 mW (from panel **A**) and 0.1 mW (from inset in panel **A**). The 1 mW 40 K spectrum resembles the 20 K spectrum besides its lower amplitude. The small signal at *g* = 1.974 (*B* ≈ 340 mT) observed in the spectra recorded at 40 K is not further addressed here due to its very low intensity. Vertical lines (dashed) are drawn to guide the eye.

**Figure 4 ijms-22-06265-f004:**
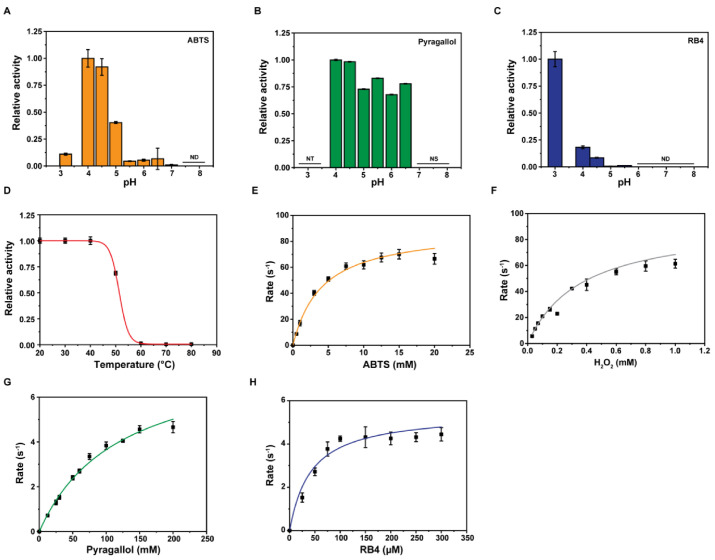
Steady-state kinetic data for *Dictyostelium* DyPA with different substrates. (**A**–**C**) Effect of pH on the activity of *Dictyostelium* DyPA. Optimum pH of *Dictyostelium* DyPA toward oxidation of ATBS, pyrogallol, and RB4. (**D**) Thermal stability of *Dictyostelium* DyPA. ND, not detected; NT, not tested, and NS, not stable (substrate was not stable at this pH). (**E**–**H**) Steady-state kinetic data for *Dictyostelium* DyPA with varying substrate concentration. The observed rates were plotted against the substrate concentrations and fitted to the Michaelis–Menten equation, and parameters are summarized in Table 2. (**E**) ABTS, (**F**) H_2_O_2_ *, (**G**) Pyrogallol, and (**H**) RB4. * 15 mM ABTS was used as a substrate for H_2_O_2_ measurements. Data are average values of 3–6 independent measurements, and bars represent the standard deviations.

**Figure 5 ijms-22-06265-f005:**
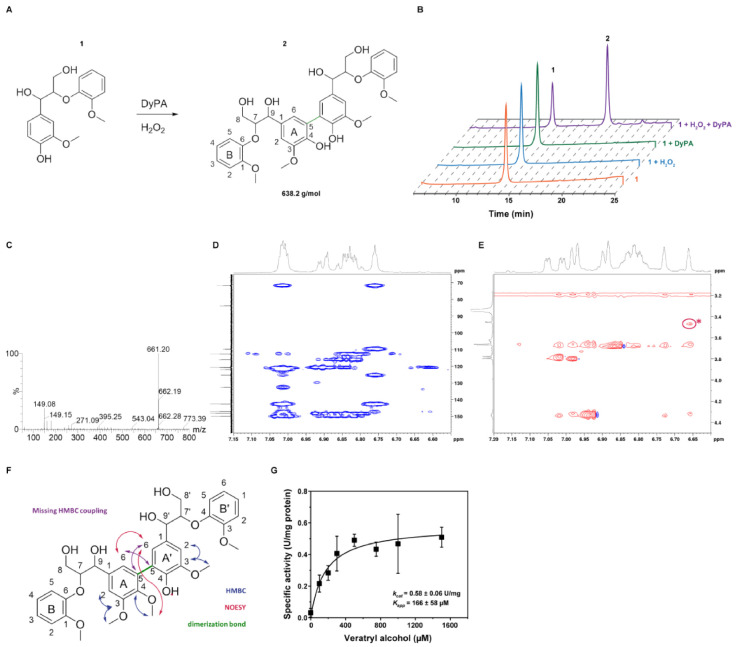
Oxidation of lignin model substrate (GGBGE) by *Dictyostelium* DyPA. (**A**) Structure of the ß-aryl ether lignin model substrate (**1**) and the proposed dimeric product (**2**). (**B**) Reverse HPLC profiles monitoring substrate turnover in the absence and presence of DyPA and H_2_O_2_. (**C**) ESI-MS of 2nd HPLC peak with an experimental m/z value of 661.20 for the Na adduct of (**2**). The calculated m/z value corresponds to 661.2261. (**D**) The structure of the catalyzed dimerization of GGBGE by *Dictyostelium* DyPA is demonstrated by a 2D HMBC experiment. This spectrum shows the proposed structure, in particular the coupling of H15 (=6.77 ppm) to A5 (=125.9 ppm). However, due to the fact that it is a symmetrical molecule, the ring systems A and A′ cannot be clearly distinguished. (**E**) The structural elucidation of the asymmetric methoxy derivative is based on NOE spectroscopy. The spectrum shows the introduced interaction between the methoxy group (d = 3.45 ppm) and C15*-H (d = 6.66 ppm), marked with a star. (**F**) The coupling scheme of GGBGE of the HMBC (blue arrows) and NOESY (red arrows) experiments. The green bond indicates the dimerization bond as the result of the *Dictyostelium* DyPA mediated dimerization. The blue methoxy group is introduced by derivatization to create the asymmetry; the NOE-signal of this asymmetry is indicated by a red star. The purple arrows indicate missing HMBC couplings for the methoxy derivative. (**G**) Oxidation of veratryl alcohol by *Dictyostelium* DyPA. Each data point corresponds to the averaged values from 3–6 independent measurements. Error bars represent the standard deviations.

**Figure 6 ijms-22-06265-f006:**
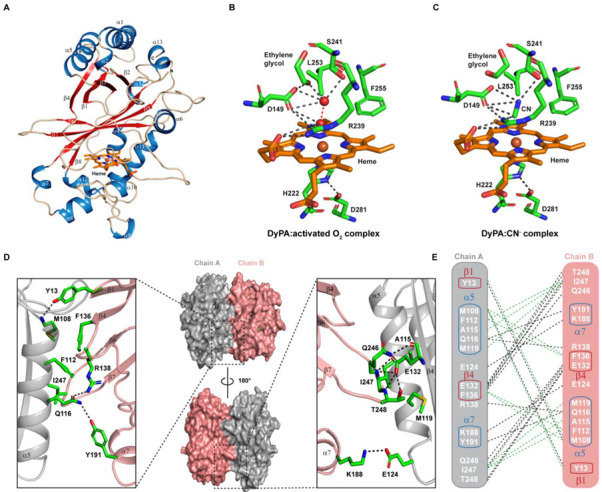
Structure of *Dictyostelium* DyPA. (**A**) Cartoon representation of the overall fold of *Dictyostelium* DyPA monomer and the heme-binding site. α-helices and β-strands are numbered in order from N to C-terminal. α-helices, β-strands, and loops are colored in sky blue, red, and wheat, respectively. Heme prosthetic group is shown as a stick model. (**B**) The heme microenvironment of *Dictyostelium* DyPA in complex with an activated oxygen molecule. The atoms of the activated oxygen molecule are shown as red spheres. (**C**) Heme microenvironment of *Dictyostelium* DyPA-CN^−^ complex. Iron, nitrogen, and oxygen atoms are colored in orange, blue, and red, respectively. Grey dashed lines indicate distances of less than 3.4 Å. (**D**) *Dictyostelium* DyPA dimer interface. *Dictyostelium* DyPA monomers are distinguished by the colors grey and salmon pink. The upper panel shows a close-up view of the interface, while the lower panel shows the view after 180° rotation. (**E**) Schematic representation of residues located at the DyPA dimer interface that contribute to the interaction between monomers A and B. Hydrogen bonds and ionic interactions are shown as black dashed lines and hydrophobic interactions as green dashed lines. In addition, the carboxyl group of Asp146 extends the hydrogen bond network between *Dictyostelium* DyPA monomers by forming an H-bond contact with Gln116 of another monomer via a water molecule. This interaction provides additional stabilization to the dimer interface of DyPA. Asp146 contributes also indirectly to the stabilization of the dimer interface by affecting the orientation of the side chain of Arg138 in the same monomer (Appendix A).

**Figure 7 ijms-22-06265-f007:**
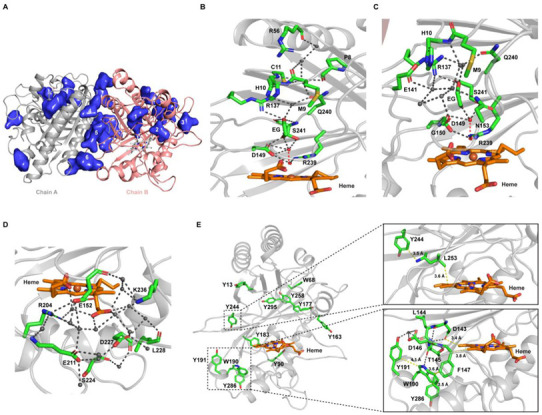
Potential substrate binding pockets, solvent-accessible channels of heme, and long-range energy transfer (LRET) sites of *Dictyostelium* DyPA. (**A**) Potential substrate-binding pockets predicted by POCASA. (**B**,**C**) Two solvent channels leading to the distal face of *Dictyostelium* DyPA heme are shown. Water molecules are shown as grey spheres. The atoms of the activated oxygen molecules are represented as red spheres. The hydrogen-bonding network leading to the active site is shown. (**D**) A conserved shallow pocket leading to the heme propionates. (**E**) Distribution of tryptophan and tyrosine residues in *Dictyostelium* DyPA. The inset displays key residues involved in proposed long-range energy transfer (LRET) sites to the heme. The first LRET site from Tyr244 leads to the distal side of heme via Leu253, while the second probable LRET site from Trp190, Tyr191, or Tyr268 leads to the proximal side of heme moiety via various possible routes. Grey dashed lines are showing hydrogen bonds.

**Figure 8 ijms-22-06265-f008:**
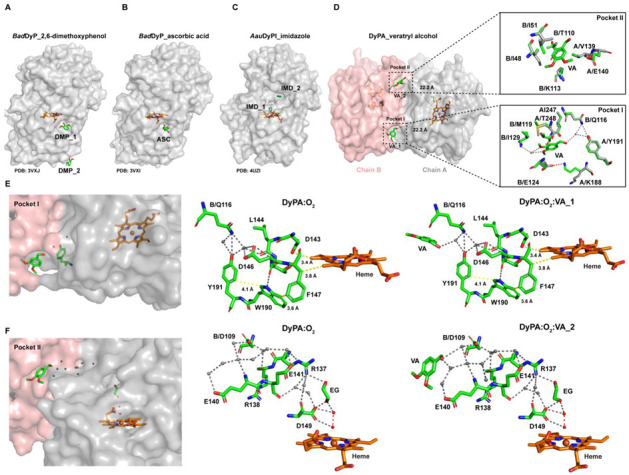
Overview of veratryl alcohol binding sites on *Dictyostelium* DyPA and comparison of substrate binding by other DyPs. (**A**) DMP binding sites in *Bad*DyP:DMP (PDB: 3VXJ) complex structure. (**B**) Ascorbic acid binding site in *Bad*DyP:ASC (PDB: 3VXI) complex structure. (**C**) Imidazole binding sites in *Aau*DyPI:IMD (PDB: 4UZI) complex structure. (**D**) Veratryl alcohol binding sites in *Dictyostelium* DyPA:O_2_:VA complex structure. The distance between the iron atom of heme moiety and the oxygen atom of the veratryl alcohol hydroxyl group is indicated by a dashed line. The lower inset shows a superimposition of substrate-free and veratryl alcohol-bound *Dictyostelium* DyPA structures for binding site I, while the upper inset shows binding site II. Carbon atoms in the substrate-free structure are represented in white and carbon atoms in the *Dictyostelium* DyPA:O_2_:VA complex are shown in green. (**E**,**F**) Proposed long-range electron transfer pathways from veratryl alcohol binding sites I and II to the heme moiety of *Dictyostelium* DyPA. Position of the *Dictyostelium* DyPA veratryl alcohol binding site I/II relative to the heme cofactor. The atoms of the activated oxygen molecules, with elongated bond distances between the oxygen atoms of 1.7 and 2.0 Å in the two monomers, are colored in red and water molecules are shown as grey spheres. Hydrogen bonds are represented as grey dashed lines.

**Table 1 ijms-22-06265-t001:** EPR parameters for simulation of the *Dictyostelium* DyPA 6K spectrum.

	***g_x_***	***g_y_***	***g_z_***	***HStrain_x_*^1^**	***HStrain_y_*^1^**	***HStrain_z_*^1^**	***E/D*^2^**	***R*^3^**	***%***
HS 1, rh	6.44	5.44	2.00	1087	183	146	0.0208	6.25%	33
HS 2, rh	6.21	5.20	1.99	343	840	89	0.0210	6.31%	44
HS 3, ax	5.97		1.97	841		111	-	-	19
LS	2.81	2.28	1.99	190	150	60	-	-	4

**^1^** Gaussian broadening assuming unresolved hyperfine couplings, full width at half maximum in MHz (see www.easyspin.org for further details, accessed on 1 April 2021). **^2^** *E/D* calculated from the absolute difference in the *g*_⊥_-values: *D/E* = (*g_x_*–*g_y_*)/48. [31] **^3^** Rhombicity (%) calculated from the absolute difference in *g*_⊥_-values: *R =* (*g_x_*–*g_y_*)/16 × 100% [42].

**Table 2 ijms-22-06265-t002:** Steady-state kinetic data for *Dictyostelium* DyPA.

Substrate	*K*_m_ (mM)	*k*_cat_ (s^−1^)	*k*_cat_/*K*_m_ (M^−1^ s^−1^)
ABTS	4.1 ± 0.4	89.9 ± 4.3	2.19 × 10^4^
H_2_O_2_ *	0.37 ± 0.063	93.65 ± 9.1	2.53 × 10^5^
Pyrogallol	119.0 ± 11.0	7.9 ± 0.4	66.38
Reactive blue 4	0.04 ± 0.01	5.2 ± 0.38	1.3 × 10^5^
Veratryl alcohol	0.166 ± 0.058	3.38 × 10^−4^ ± 0.35 × 10^−4^	2.03

1 mM H_2_O_2_ was used as a co-substrate for *K*_m_ determination. * 15 mM ABTS was used as substrate.

## Data Availability

Atomic coordinates and structure factors have been deposited in the Protein Data Bank with accession codes 7O9J (*Dictyostelium* DyPA:O_2_ complex at 1.7 Å), 7O9L (*Dictyostelium* DyPA:CN^−^ complex at 1.85 Å), and 7ODZ (*Dictyostelium* DyPA:O_2_:veratryl alcohol complex at 1.6 Å).

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
