# Peer review of "Structural and Biochemical Characterization of a Dye-Decolorizing Peroxidase from Dictyostelium discoideum"

_ijms, 2021, doi:10.3390/ijms22126265_

Round 1

Reviewer 1 Report

The paper by Rai et al reports a broad characterization of the Dictyostelium discoideum peroxidase (Dictyostelium DyPA), using different spectroscopic/hydrodynamic/kinetic techniques. This work also includes three different X-ray structures of DyPA (free and bound to KCN and veratryl alcohol (VA)). Whereas there are many different homolog structures deposited at the PDB (e.g. Klebsiella, Enterobacter, Escherichia, Rhodococcus…), the structural characterization per se, and the structural analysis of the different structures provide interesting clues on the reaction mechanism of these enzymes. Specifically, and as far as I know, whereas the location of surface spots has been reported for other peroxidases/other families (such as lignin peroxidases), this might be the first time that surface/interace binding has been reported for this class of peroxidases. Thus, the location of the putative substrates on different pockets might be of interest for researchers in the field, to further gain insights into a putative LRET mechanism. On the other hand, there are some concerns that should be addressed.

Major concerns:

  • Dictyostelium DyPA is purified showing a low amount of heme group according to the Soret peak (lines 137-141). The authors state that the enzyme was reconstituted for some of the experiments (line 141). Please, clarify how reconstitution was conducted, since their previous work on the purification of the enzyme (ref 15) did not provide this methodology, and it is quite relevant. Also, further specify if the reconstituted enzyme was used for most of the experiments. In the same sense, were the structures obtained with the reconstituted enzyme? If not, what was the occupancy of the heme group?

  • Line 318. The authors provide an apparent “melting temperature” for DypA. However, Fig4D shows a single experimental data different to 100% or 0% activity (that at 50ºC). Thus, this thermal midpoint is unreliable. If no more data is included to calculate this apparent thermal midpoint, please rephrase/soften this result, even when the conclusion will be the same as that reported (lines 328-329)

  • The authors propose a dimeric lignin adduct (specie 2, Fig 5A), but the results do not totally support this specie (theoretical mass of "2"= 638.2 gr/mol; ESI-MS showed a m/z of 661.46 (Figure 5C). Whereas I mostly agree that a dimeric species could appear, since the results do not totally agree with this hypothesis, I suggest moving this part to the SI, indicating that a specie similar to “2” could be the product of the reaction, based on the NMR and ESI-MS results.

  • I have no access to the PDB structures, nor to the mtz-files reported in the paper. Thus, and without being able to check the binding-site fitting conducted by the authors, there are some concerns on the binding of VA to the surface/interace of DyPA. Firstly, the authors suggest the binding of VA to “pockets” 4 and 6, and based on previous works on different peroxidases (see below), they hypothesize on a plausible LRET mechanism. I find it necessary to show/compare the distances between the ligand-binding sites in DyPA and other peroxidasesfor which LRET mechanisms were previously proposed, since the distance to the heme group is larger than 20A in both cases; this distance is quite lower in some of the enzymes shown in Figure 8.

  • I partly agree with the proposed LRET hypothesis: it has been described elsewhere (a similar scenario was proposed for Auricularia auricula-judae peroxidase, which might help the authors to further support their hypothesis; refs 43, 56, 58). Furthermore, the EPR support the formation of radical Trp and perhaps, Tyr residues. However, to ascertain the role of these residues, site-directed mutagenesis should be conducted. Since I do not find strictly necessary to include further experiments, as the LRET hypothesis is widely described elsewhere, I think that the authors could soften the statements on the plausible LRET mechanism, using further references where similar/counterpart Trp/Tyr residues have been proved to form radical species, or even theoretical studies on this mechanism (e.g., Acebes et al., 2017. J Phys Chem B. 121(16):3946-3954; Ruiz-Dueñas et al., 2009 J Exp Bot. 60(2):441-52; Romero et al., 2019. Comput Struct Biotechnol J. 17:1066-1074)

  • Line 526: please, specify a reference where the presence of a tyrosine residue for LRET mechanism has been proposed/shown (such as PDB 4UZI, ref 56).

  • Other minor concerns.

- As I mentioned previously, I had no access to the PDB nor mtz files. Thus, it is difficult for me to evaluate the binding of the different ligands, and specifically for VA, which is the ligand providing the novelties of this work. If obtaining the different PDBs and structures factors is not possible for additional review, please include at least an OMIT map for the three ligands, representing it to 1.0-1.5 sigma, at least for review purpose.

- The authors could shorten the POCASA results (lines 496-514), and/or move them partially to the SI; this software supports the appearance of VA on the proposed pockets, but the experimental results are the key to support the LRET hypothesis. In this sense, I would also recommend to firstly introduce the appearance of the VA molecules based on the experimental X-Ray results, and later on, supporting substrate binding with this tool.

- Change Dictosyleum to italics through the whole manuscript.

- Please, specify whether the crystals obtained by co-crystallization with KCN and VA were obtained in the same condition as the free form.

-Since formation of Compound I has been recently studied in detail for Streptomyces peroxidase, the authors should comment on this paper and the relationship with the results presented in their work (Lučić et al., 2020 Angew Chem Int Ed Engl. 59(48):21656-21662. )

Reviewer 2 Report

Amrita et al. analyzed the function of a dye decolorizing peroxidase from Dictyostelium discoideum using extensive experimental tools. The manuscript reveals a vast array of DyPA features and functions from various experimental tools. The results of this study are considered to be used as good literature in related fields. Before the manuscript is recommended for publication on IJMS, I suggest to improve the contents below.

  1. In the DyPA:KCN crystal structure, isn't the N atom of the CN molecule supposed to be bonded to the Fe of heme? The authors have to provide an electron density map for CN bonds and comment on the orientation of the CN molecule. Meanwhile, what is the B-factor or occuancy for heme or CN?
  2. I suggest that the authors add a conclusion and put together the highlights of this study's findings here. This would be good to remind readers of the important results of the manuscript.

  3. Are the two molecules in the asymmetric unit similar in length of interaction at Heme or other major active sites? The author describes any similarity or differences between these two structures.

Minor

  1. line 44-50: Authors should add references.

  2. line 117-118:"Sequence alignment was performed using Clustal Omega." Author should move this sentence to the "Materials and methods" section.

  3. line 125-126: "Sedimentation coefficient distributions were converted to 12 mm path length for better comparison." Author should move this sentence to the "Materials and methods" section.

  4. line 131-132: "The buffer used contains 50 mM Tris-HCl pH 8.0 and 150 mM NaCl." Author should move this sentence to the "Materials and methods" section.

  5. Go to line 133-134: "KCN. The buffer used contains 50 mM Tris-HCl pH 8.0 and 150 mM NaCl." Author should move this sentence to the "Materials and methods" section.

  6. line 235-236: "The simulation parameters are given in Table II." Author should move this sentence to the text, not figure caption.

  7. line 387: "The absorbance of substrate and product was detected at 254 nm." Author should move this sentence to the "Materials and methods" section.

  8. Figure 6E: It is difficult to distinguish each lines. Author should increase the size of Figure 6E to see easily (maybe it is appropriate to move to supplementary figure).

  9. line 471: what is "(4)"

Figure S1: Author should add the sequence accession number and information about the strain.

Figure S4: It is suggested to concise the figure caption. (E.g., Structural alignment of Dictyostelium DyPA (gray) with (A) Escherichia coli O157 EfeB (blue; class A), (B) Escherichia coli O157 YfeX (blue; class B), (C) Klebsiella pneumoniae KpDyP (blue; class B) ....)

Figure S5 and S6: If the two molecules in the asymmetric unit have different configurations, it should be added them.

Figure S7: Please add the length of the interaction cutoff.

Round 2

Reviewer 2 Report

The author has addressed all my concerns, and the revised manuscript has been improved. I recommend publication of this manuscript after correction of below issues,

1. The references do not match the journal format. (e.g. Pfanzagl V, Nys K, Bellei M, Michlits H, Mlynek G, Battistuzzi G, et al. Roles of distal aspartate and arginine of B-class dye-decolorizing peroxidase in heterolytic hydrogen peroxide cleavage. J Biol Chem. 2018;293(38):14823-38. Epub 2018/08/04. doi: 10.1074/jbc.RA118.004773. PubMed PMID: 30072383; PubMed Central PMCID: PMCPMC6153280.) Correction is required accordingly. 

2. In Table S1,
- "Wilson B-factor" should be "Wilson B-factor  (Å2)"

- "2.3 ( 27.9)" should be "2.3 (27.9)"

- I personally think that "Rsimga" and "Rint" are not the notation used in general crystallographic papers. Accordingly, for the general reader, it is suggested to further explain "Rsimga" and "Rint" as footnotes.